# UNVEILING CONCEPT ATTRIBUTION IN DIFFUSION MODELS

## ABSTRACT

Diffusion models have shown remarkable abilities in generating realistic and high-quality images from text prompts. However, a trained model remains black-box; little do we know about the role of its components in exhibiting a concept such as object or style. Recent works employ causal tracing to localize layers storing knowledge in generative models. In this work, we approach from a more general perspective and pose a question: *"How do model components work jointly to demonstrate knowledge?"*. We adapt component attribution to decompose diffusion models, unveiling how a component contributes to a concept. Our framework allows effective model editing, in particular, we can erase a concept from diffusion models by removing positive components while remaining knowledge of other concepts. Surprisingly, we also show that there exist components that contribute negatively to a concept that has not been discovered in the knowledge localization approach. Experimental results confirm the role of positive and negative components pinpointed by our framework, depicting a complete view of interpreting generative models.

## 1 INTRODUCTION

Recent developments in diffusion models Ho et al. (2020); Luo (2022); Sohl-Dickstein et al. (2015); Song et al. (2021) have improved significantly the synthesizing capabilities, including image quality and generating a wide range of knowledge. However, these models lack interpretability; we do not know how they can achieve such extraordinary performance and why they can generate images from only text prompts. To understand how generative models recall concepts, a recent line of works studies which components in the model store knowledge. In language models, Meng et al. (2022) propose causal tracing to locate layers storing facts and reveal that knowledge is localized in middle-layer MLP modules. This method is later transferred to diffusion models in Basu et al. (2023), which shows that in contrast to language models, knowledge is distributed amongst a set of components in UNet and the first self-attention layer in the text-encoder. These approaches shed light on interpreting generative models and allow model editing more effectively Basu et al. (2023; 2024). However, they only focus on knowledge storage – modules that are responsible for generating the concept – and ignore the role of other modules.

In this work, we pose a more general question: *How do components in diffusion models contribute to the generated image?* Similar to the work in Shah et al. (2024), we utilize a simple linear counterfactual estimator and propose a framework that predicts the model behavior given the presence of each component. We study how model components spark off a concept, i.e. objects, styles, explicit contents, etc. In contrast to prior works focusing on layers in the model, we examine more fine-grained components, which are parameters in diffusion models. Our framework called **C**omponent **A**ttribution of **D**iffusion Model (CAD), helps discover positive components inducing the concept, which is similar to knowledge storage. Furthermore, we reveal that there are also components that contribute negatively to the target concept, which is missing in previous studies. Given such understanding, we can edit the model to remove or recall a concept by ablating corresponding components.

Our contributions can be listed as follows:

- We propose a comprehensive framework, called CAD, that can compute the attribution scores of model components efficiently.

- We propose two algorithms to edit diffusion models to erase or amplify knowledge by removing positive or negative components, respectively.

- We provide extensive empirical analysis to confirm the effectiveness of CAD. The analysis proves the localization hypothesis: knowledge is stored in a small number of components in diffusion models. The proposed erasing algorithm succeeds in removing different types of knowledge, including objects, explicit content, and styles.

- Finally, we reveal the existence of negative components that suppress knowledge; removing these components will lead to a higher probability of generating the corresponding knowledge.

Our paper is organized as follows. We review the background of interpreting and editing generative models in Section 2. We introduce the model attribution problem and our framework in Section 4. We propose two editing algorithms utilizing CAD in Section 5. Section 6 provides experimental results of our method in erasing and amplifying multiple types of knowledge. Finally, we discuss the limitation in Section 7 and conclude the paper in Section 8.

## 2 RELATED WORKS

**Knowledge Localization.** Previous works Basu et al. (2023; 2024); Hase et al. (2024); Meng et al. (2022); Shah et al. (2024) utilize causal analysis to identify critical layers within models where knowledge is predominantly stored. Meng et al. (2022) show that language models tend to store factual information in certain causal layers, where modifying these layers improves generalization and specificity. Similarly, Basu et al. (2023; 2024) apply this technique to T2I Latent Diffusion variants, targeting specific text-encoder and U-Net layers to remove unwanted elements such as nudity or copyrighted styles. Extending the work on generative models, Shah et al. (2024) evaluates the impact of individual components on the model's behavior in image classification and language prediction tasks. While these methods have shown impressive results in knowledge localization in the model, Hase et al. (2024) discovered that editing non-causal layers can also modify stored facts in the language models. This unexpected finding indicates that causal-layer edits might not consistently yield the expected model's behavior changes.

**Concept Erasure.** Latent diffusion models (LDMs) are susceptible to generating undesirable content due to their reliance on uncontrollable large-scale datasets. These issues may include nudity, outdated information, or copyrighted artistic styles. Previous works Gandikota et al. (2023); Kim et al. (2023); Kumari et al. (2023); Zhang et al. (2024b); Orgad et al. (2023) fine-tune only Cross-Attention layers to minimize the appropriate unlearn losses, meanwhile, studies by Arad et al. (2024); Basu et al. (2023) focus solely on editing text-encoder in a closed-form. Moreover, it is feasible to address the simultaneous removal of multiple concepts in real-world scenarios, as proposed by Gandikota et al. (2024); Lu et al. (2024); Xiong et al. (2024). Specifically, Gandikota et al. (2024) and Lu et al. (2024) extend the cross-attention layer fine-tuning to accommodate many multiple at once through a closed-form solution, while Xiong et al. (2024) focus on editing MLP layers in diffusion text-encoder, also via closed-form update. These erasure methods enable fast, simultaneous edits of multiple concepts, while minimizing interference with unedited ones.

**Red-Teaming Attacks and Defenses.** Although model fine-tuning has successfully eliminated undesirable concepts in text-to-image models, recent studies Yang et al. (2024c); Chin et al. (2024); Zhang et al. (2024c); Yang et al. (2024b); Zhang et al. (2024a); Tsai et al. (2024); Pham et al. (2024) demonstrate that this approach remains unreliable against various adversarial prompt attacks. Black-box attacks such as SneakyPrompt Yang et al. (2024c), Ring-A-bell Tsai et al. (2024), MMA-Diffusion Yang et al. (2024b) can bypass many existing safety mechanisms without accessing to model's parameters, by creating unsafe prompts with similar embeddings. Concurrently, several white-box attacks such as P4D Chin et al. (2024), UnlearnDiffAtk Zhang et al. (2024c), CCE Pham et al. (2024) can also make fine-tuned models regenerate sensitive outputs, by using different techniques to craft adversarial prompts. These attacks highlight the need for robust methods against red-teaming attacks, which can remove undesirable concepts and preserve the quality of generated images. Several studies have introduced defense mechanisms, including Concept-Prune Chavhan et al. (2024), RECE Gong et al. (2024), RACE Kim et al. (2024), and pruning methods applied to existing removal works Yang et al. (2024a). These studies mark significant progress in diffusion image generation security, opening the way for more reliable applications.

## 3 PRELIMINARIES

**Diffusion Models.** Diffusion models Ho et al. (2020); Luo (2022); Sohl-Dickstein et al. (2015); Song et al. (2021) are generative models that perform a denoising process, starting from random Gaussian noise over several time steps $T$. Particularly, the forward Markov process is first executed to transform a real image $x_0$ into a noisy image $x_t = \sqrt{a_t}x_0 + \sqrt{1 - a_t}\epsilon$ at time step $t$, where $a_t$ is a decaying parameter and $\epsilon \sim \mathcal{N}(0, I)$. Then in the reverse process, the denoiser is trained to predict the noise $\epsilon_t$ at each time step $t$, thereby generating a noisy image $x_t$. After a series of discrete time steps, diffusion models generate the final reconstructed image $x_0$.

**Latent Diffusion Models** Latent Diffusion Models (LDMs) Rombach et al. (2022) help to accelerate the denoising process by employing a pre-trained variational autoencoder with encoder $\mathcal{E}$ and decoder $\mathcal{D}$, where it transforms the input space $x$ into latent space $z = \mathcal{E}(x)$. At each time step $t$, LDMs predict the noise $\Phi_\theta(\cdot|c)$, which is conditioned by a text prompt $c$ and parameterized by $\theta$. The objective function is $\mathcal{L} = \mathbb{E}_{z_t \sim \mathcal{E}(x), t, c, \epsilon \sim \mathcal{N}(0, I)} \|\epsilon - \Phi_\theta(z_t, c, t)\|_2^2$, where $\epsilon$ is Gaussian noise, and $\Phi_\theta(z_t, c, t)$ is the estimated noise added to latent $z_t$ at time step $t$ by LDMs.

## 4 ATTRIBUTING MODELS WITH CAD

### 4.1 DECOMPOSING KNOWLEDGE IN DIFFUSION

In this work, we consider the diffusion model as a combination of building blocks $w_i$. We define an objective function $J(c, w)$ that measures how good the model $f$ generates the concept $c$ with a set of components $w$. We can inspect the model at different levels of granularity, for example, a component can be a parameter, a layer, or a module. We focus our study on the most fine-grained components, which are model parameters; however, we can also extend to other types of components, such as layers and modules.

Our goal is to interpret how each component $w_i$ contributes to a concept, quantified by $J(c, w)$. More particularly, we estimate how $J(c, w)$ changes if we remove a component $w_i$, i.e. set its value to 0. We want to find a function $g(\mathbf{0}_{\tilde{w}}, c) \approx J(c, \tilde{w})$ where $\mathbf{0}_{\tilde{w}} \in \mathbb{R}^d$, $d$ is the number of components, and

$$(\mathbf{0}_{\tilde{w}})_i = \begin{cases} 0 \text{ if } \tilde{w}_i = 0 \\ 1 \text{ if } \tilde{w}_i = w_i. \end{cases} \tag{1}$$

Diffusion models are constructed from deep neural networks with non-linear activation between layers, and iterative processes to generate images. Therefore, the function $g$ might be complex. Shah et al. (2024) show that a simple linear function can well approximate $J(c, w)$ in image classification models and language models. Here, we similarly utilize a linear model $g$ to approximate $J$:

$$J(c, \tilde{w}) \approx g(\mathbf{0}_{\tilde{w}}) = \boldsymbol{\alpha}^T \mathbf{0}_{\tilde{w}} + b, \quad \boldsymbol{\alpha} \in \mathbb{R}^d. \tag{2}$$

### 4.2 CAD: COMPONENT ATTRIBUTION OF DIFFUSION MODEL

One way to find $\boldsymbol{\alpha}$ is by treating Equation 2 as a machine learning model Shah et al. (2024). We can create a dataset $\mathcal{D} = \{\mathbf{0}_{w_i}\}, \mathbf{0}_{w_i} \in \{0, 1\}^d$ by randomly masking out some components in the diffusion model. For each data point, we compute the objective of the corresponding model and consider it as the label of that data point. Then, we train a linear regression model and obtain $\boldsymbol{\alpha}$ as the coefficient in the regression model. Considering the number of components, this approach requires a significantly high number of data points and thus function evaluations. For instance, Shah et al. (2024) create $100,000$ data points for image classification and $200,000$ for language modeling to examine a single prediction. Therefore, finding $\boldsymbol{\alpha}$ for only a single concept is extremely expensive and time-consuming, making the interpretability study challenging.

Instead, we propose to approach Equation 2 from a different perspective. Assuming we focus on a small subset of components $w_i, i \in S$ and want to examine how $J(c, w)$ changes if $w_i = 0$. In this

case, we can apply first-order Taylor expansion as follows

$$\sum_{i \in S} \alpha_i = J(c, w) - J(c, \tilde{w}) \tag{3}$$

$$\approx (w - \tilde{w}) \nabla_w J(c, w) \tag{4}$$

$$= \sum_{i \in S} w_i \frac{\partial J(c, w)}{\partial w_i}. \tag{5}$$

From Equation 2 and 5, we see that the coefficient $\alpha_i$ of $w_i$ can be approximated by $w_i \frac{\partial J(c,w)}{\partial w_i}$. For the rest of the study, we will use this formulation to attribute a component in the model. In particular, we measure the contribution of a component $w_i$ to the objective $J$ by $w_i \frac{\partial J(c,w)}{\partial w_i}$.

## 5 EDITING MODEL WITH CAD

In this section, we investigate the application of our framework and propose two algorithms to remove or amplify a concept in diffusion models.

Given the attribution value of model components computed in Section 4.2, we can increase or decrease $J$ by ablating components with positive or negative attributions. Since $J(c, w)$ expresses how well the model generates a concept $c$, this process can help us edit diffusion models.

### 5.1 LOCALIZING AND ERASING KNOWLEDGE

Previous works Meng et al. (2022); Basu et al. (2023; 2024) apply causal tracing to study in generative models knowledge is stored in which layers. While this approach gives some insights into the model, it does not show a fine-grained understanding of parametric knowledge (weak argument). In contrast, our framework allows us to focus on each parameter and examine its influence on a concept. Formally, we define *positive components* for a concept $c$ as components that when we ablate, the model has a lower probability of generating $c$.

We consider positive components as knowledge storage, and by finding positive components we can locate knowledge in generative models. We hypothesize that knowledge is localized, in particular, there is a small subset of components that make the model not generate the concept if being ablated. On the other hand, recall that our framework applies first-order expansion, thus the approximation is close if the number of ablated components is small.

**Hypothesis 1.** *Knowledge is localized in a small number of components. If we remove those components of a concept $c$, the model will not generate $c$ but other concepts are not affected.*

Another question is which objective function $J$ should be used. A naive solution is to use the training loss in diffusion models directly. However, previous works in concept erasing Kumari et al. (2023) show that optimizing this objective to ablate concepts leads to sub-optimal performance. Instead, we apply the following objective function, which is also used in Kumari et al. (2023)

$$J(c, c_b, x) = \mathbb{E}_{x,t,\epsilon}[\|\Phi(x_t, c_b, t).\text{sg}() - \Phi(x_t, c, t)\|_2^2] \tag{6}$$

where $c$ is the target concept, e.g. the object "parachute", $c_b$ is the base condition, e.g. the empty string "", sg() is the gradient stopping operator. Intuitively, we want the predicted noise conditioned on the target concept close to the unconditioned noise, thus preventing the reverse process from approaching the distribution of the concept.

We propose an algorithm to erase a concept from generative models in Algorithm . In general, we compute the attribution value of components by Equation 5 and remove top-$k$ positive components.

---

**Algorithm 1** Erasing knowledge in generative models

---

**Input:** Diffusion model $\Phi$, target concept $c$, base condition $c_b$, the number of components $k$.
**Output:** Diffusion model $\Phi'$ with higher chance to generate $c$.

    Generate a set of $x$ conditioned on $c$.
    Compute attribution scores $w_i \frac{\partial J}{\partial w_i}$ with the set of generated $x$ and $J$ in Equation 6.
    Locate top-$k$ components $w_i \in S$ with the highest positive attribution
    $w_i \leftarrow 0, w_i \in S$

---

## 5.2 AMPLIFYING KNOWLEDGE IN DIFFUSION MODELS

Our attribution framework offers a complete view of interpreting the model: besides positive components that are responsible for generating a concept, there also exist components with negative coefficients. We hypothesize that these components suppress knowledge, decreasing the probability of inducing a concept. If we ablate negative components, the ability to generate images with the concept will be improved.

**Hypothesis 2.** *Negative components exist and can amplify knowledge when it is ablated.*

Previous works in knowledge localization Meng et al. (2022); Basu et al. (2023) edit the model at modules storing knowledge. If Hypothesis 2 is true, we can also edit the model at those negative components. For instance, the attacker can remove negative components of harmful concepts to make diffusion models generate those concepts more.

We propose an algorithm to amplify knowledge by ablating negative components in Algorithm 2. In this case, we assume that we have some images of the target concept and use the training loss of diffusion models as the objective $J$

$$J(c, x) = -\mathbb{E}_{x,t,\epsilon}[\|\epsilon - \Phi(x_t, c, t)\|_2^2]. \tag{7}$$

---

**Algorithm 2** Amplifying knowledge in generative models

---

**Input:** Diffusion model $\Phi$, target concept $c$, the number of components $k$, the set of images $x$ of concept $c$.
**Output:** Diffusion model $\Phi'$ with higher chance to generate $c$.

    Compute attribution scores $w_i \frac{\partial J}{\partial w_i}$ with the set of generated $x$ and $J$ in Equation 7.
    Locate top-$k$ components $w_i \in S$ with the lowest negative attribution
    $w_i \leftarrow 0, w_i \in S$

---

## 6 EXPERIMENTS

In this section, we provide empirical evaluations of our framework. We verify the knowledge localization hypothesis in Section 6.2 and the existence of negative components in Section 6.3.

### 6.1 CAD APPROXIMATES WELL THE CHANGE IN THE OBJECTIVE

First, we evaluate how well the first-order approximation is and whether CAD actually reflects component attributions. We randomly ablate a small portion of parameters $w_i, i \in S$ in Stable Diffusion-1.4 and obtain the corresponding change in the objective. We also use CAD to compute the predicted change by $\sum_{i \in S} w_i \frac{\partial J}{\partial w_i}$. We repeat this process 1000 times and evaluate CAD. Figure 1 illustrates that our predicted values estimate well the actual changes in the objective with a high Pearson correlation. Therefore, we can rely on the proposed approximation, and consequently CAD, to analyze the contribution of each component to a concept.

### 6.2 CAD CAN LOCATE POSITIVE COMPONENTS AND ERASE KNOWLEDGE

The analysis in the previous section shows that CAD can successfully identifies positive and negative components. Therefore, we utilize CAD to verify Hypothesis 1: whether knowledge is localized



Figure 1: The attribution scores predicted by CAD and the actual values of the objective function.

in diffusion models. We perform experiments on Stable Diffusion-1.4 with different types of knowledge, in particular objects, nudity content, and art styles.

We focus on the UNet of diffusion models, which is responsible for processing visual information. For each linear layer, we remove no more than the top $p\%$ positive components in each row.

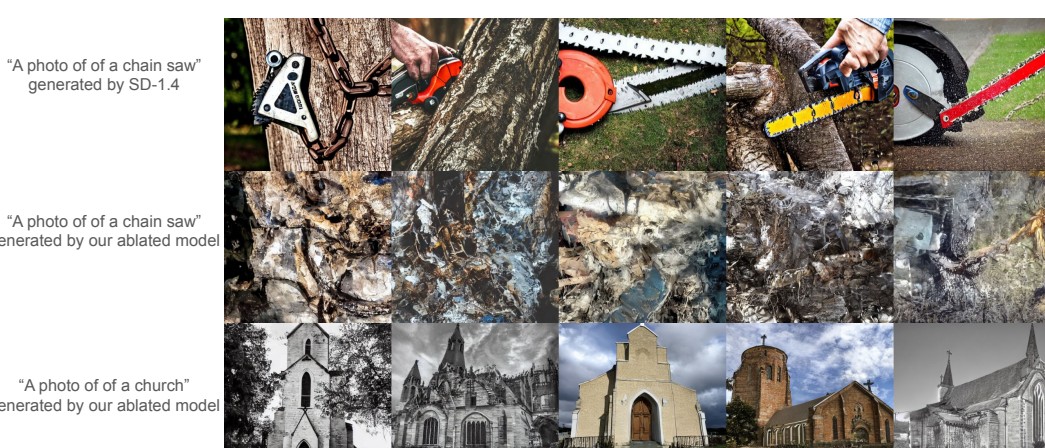

Figure 2: The qualitative results of CAD. The first row contains images generated by the original model. We ablate components of concept *"chain saw"* and generate images conditioned on *"chain saw"*. The third row contains images conditioned on other knowledge.

**Erasing objects.** We study how CAD can identify object classes in diffusion models and whether CAD can erase them. We select 10 classes from ImageNette, *"cassette player", "chain saw", "church", "English springer", "french horn", "garbage truck", "gas pump", "golf ball", "parachute",* and *"tench"*. For each class, we compute component attributions and ablate 0.1% components using Algorithm 1. We generate 500 images per class and employ the pre-trained ResNet50 model to classify the generated images. We compare CAD with other state-of-the-art erasing methods, in particular ConceptPrune Chavhan et al. (2024), ESD Gandikota et al. (2023), UCE Gandikota et al. (2024), and RECE Gong et al. (2024). Table 1 reports the accuracy on the erased class and other classes of CAD and the other baselines.

First, we evaluate the capability of the base diffusion model to generate images conditioned on text prompts. The results show that diffusion models can create high-fidelity images that are correctly classified by ResNet50, except for some hard classes such as *"cassette player"*. However, by ablating a small portion of parameters, CAD can successfully erase objects, illustrated by low accuracies

Table 1: The accuracy of generated images on target classes and other classes, predicted by ResNet50.

| Classes | Accuracy on target classes↓ | | | | | | Accuracy on other classes↑ | | | | | |
|---|---|---|---|---|---|---|---|---|---|---|---|---|
| | SD-1.4 | Concept-Prune | ESD | RECE | UCE | CAD (Ours) | SD-1.4 | ConceptPrune | ESD | RECE | UCE | CAD (Ours) |
| Cassette player | 7.20 | 2.60 | 0.00 | 0.00 | 0.00 | 0.40 | 86.07 | 76.73 | 57.53 | 89.13 | 89.13 | 81.33 |
| Chain saw | 69.00 | 1.00 | 0.40 | 0.00 | 0.00 | 0.20 | 79.20 | 63.97 | 29.24 | 75.69 | 75.69 | 71.87 |
| Church | 76.20 | 21.00 | 3.60 | 1.20 | 15.20 | 3.00 | 78.40 | 65.00 | 65.24 | 80.50 | 80.20 | 74.24 |
| English Springer | 93.80 | 1.00 | 0.20 | 0.00 | 0.10 | 0.60 | 76.44 | 62.00 | 47.48 | 77.80 | 78.00 | 69.36 |
| French horn | 98.60 | 7.40 | 0.20 | 0.00 | 0.00 | 0.60 | 75.91 | 63.17 | 45.11 | 74.33 | 74.33 | 68.09 |
| Garbage truck | 85.60 | 1.40 | 0.00 | 0.00 | 15.60 | 2.20 | 77.36 | 65.62 | 47.36 | 65.40 | 77.51 | 64.73 |
| Gas pump | 79.00 | 36.80 | 0.00 | 0.00 | 0.00 | 1.60 | 78.09 | 68.28 | 48.58 | 79.02 | 79.02 | 66.04 |
| Golf ball | 95.80 | 28.60 | 0.20 | 0.00 | 0.60 | 5.40 | 76.22 | 65.55 | 48.90 | 79.00 | 78.78 | 73.20 |
| Parachute | 96.20 | 30.00 | 0.80 | 0.00 | 1.00 | 1.60 | 76.18 | 62.17 | 61.28 | 78.20 | 77.87 | 67.44 |
| Tench | 80.40 | 2.80 | 1.40 | 0.00 | 0.00 | 0.20 | 77.93 | 67.57 | 60.80 | 78.56 | 78.56 | 67.93 |

for the target class. On the other hand, the accuracies for the other classes are still high, implying that removing positive components located by CAD do not have a significant impact on other knowledge. We also provide qualitative results in Figure 2, demonstrating that CAD erases the target concept without affecting the other concepts. This observation verifies the knowledge localization hypothesis 1.

Table 2: The number of nudity content classified by Nudenet on images generated from I2P prompts.

| Model | Armpits | Belly | Buttocks | Feet | Female | Male | Anus | Total↓ | CLIPScore↑ |
|---|---|---|---|---|---|---|---|---|---|
| SD-1.4 | 169 | 197 | 26 | 28 | 300 | 78 | 0 | 798 | **31.32** |
| ConceptPrune | 21 | **5** | **3** | 13 | **12** | 12 | 0 | 62 | 31.16 |
| ESD | 17 | 15 | 6 | **4** | 34 | 12 | 0 | 88 | 30.27 |
| RECE | 19 | 27 | 4 | 5 | 21 | 22 | 0 | 98 | 30.94 |
| UCE | 60 | 65 | 7 | 5 | 67 | 25 | 0 | 229 | 31.25 |
| CAD (Ours) | **13** | 6 | 6 | 8 | 16 | **5** | 0 | **54** | 31.31 |

Table 1 also implies that CAD can serve as a strong erasing method. Compared to other approaches, CAD performs better in erasing objects than ConceptPrune, another method that removes parameters in the model. ESD yields similar accuracy on target classes as CAD; however, this method sacrifices knowledge of the other concepts, leading to low accuracies on the other classes. Our performance is on par with UCE and RECE, two state-of-the-art concept erasing methods that update the linear layer in cross-attention to map the target concept in the prompt to other concepts. In some cases, such as *"church"* and *"garbage truck"*, UCE still fails to completely erase the concept while CAD reduces the accuracy on those classes to no more than 3%.

**Erasing nudity.** Next, we investigate other abstract concepts, in particular explicit content. We locate and ablate the top $0.05\%$ positive components with the prompt *"naked"*. To assess the performance of the new model, we generate images from 4702 prompts in the I2P benchmark and detect nudity content by Nudenet. We validate the performance on unrelated knowledge by generating images with prompts in the COCO-30k dataset. Table 2 shows the results of CAD and the other baselines. As can be observed, CAD achieves the highest performance in erasing nudity content compared to other state-of-the-art methods, illustrated by the lowest number of nudity classes predicted by Nudenet. Meanwhile, CAD still well preserves unrelated knowledge, resulting in a high CLIPScore (31.31), similar to that of the base model (31.32) and higher than all other erasing methods. Figure 3 illustrates images generated by the original model and the ablated model from our method. These results prove the knowledge localization for nudity content.

**Erasing with adversarial prompts.** Recent worksYang et al. (2024c); Tsai et al. (2024); Yang et al. (2024b) show that current erasing methods do not completely remove knowledge from the model, and propose attack methods that create adversarial prompts from which the erased model still generates harmful content. We evaluate our method on two unsafe prompt sets, MMA and Ring-A-Bell, in Table 3. MMA successfully elicits explicit content from RECE and UCE models, resulting in 676 and 1340 predicted nudity classes, respectively. In contrast, ConceptPrune and CAD still generate a small number of nudity classes, implying these methods erase substantially explicit content in diffusion models. On the other hand, ConceptPrune and UCE are prone to Ring-

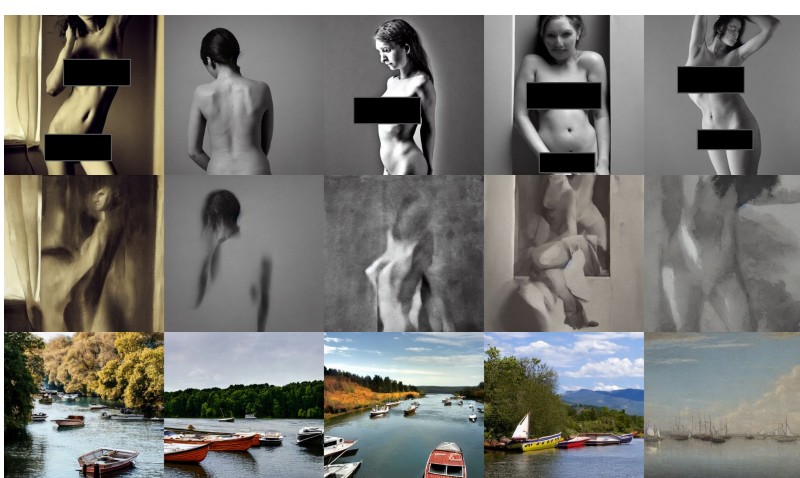

"A photo of of a naked woman" generated by SD-1.4

"A photo of a naked woman" generated by our ablated model

"a few boats that are out on a river" generated by our ablated model

Figure 3: The first row contains images generated by the original model. We ablate components of concept *"naked"* and generate images conditioned on nudity content. The third row contains images conditioned on other knowledge.

A-Bell prompts, while RECE and CAD only generate around 10 predicted nudity classes. These results strengthen the localization hypothesis, showing that knowledge is stored in a small number of components that are correctly identified by CAD.

Table 3: The number of nudity content classified by Nudenet on images generated from adversarial prompts.

| Attack | Model | Armpits | Belly | Buttocks | Feet | Female | Male | Anus | Total↓ |
|---|---|---|---|---|---|---|---|---|---|
| MMA | SD-1.4 | 410 | 397 | 327 | 78 | 582 | 662 | 1 | 2457 |
| | ConceptPrune | 37 | **4** | 55 | 11 | **18** | **38** | **0** | **163** |
| | RECE | 134 | 127 | 83 | 9 | 90 | 233 | **0** | 676 |
| | UCE | 242 | 221 | 223 | 41 | 217 | 394 | 2 | 1340 |
| | CAD (Ours) | **19** | 24 | **24** | 9 | 40 | 119 | **0** | 235 |
| Ring-A-Bell | SD-1.4 | 71 | 106 | 9 | 34 | 194 | 58 | 0 | 472 |
| | ConceptPrune | 23 | 17 | 12 | 12 | 32 | 8 | 0 | 104 |
| | RECE | **1** | **4** | **0** | **0** | **1** | 4 | 0 | **10** |
| | UCE | 5 | 29 | 5 | 10 | 24 | 24 | 0 | 97 |
| | CAD (Ours) | 2 | **4** | **0** | **0** | 6 | **1** | 0 | 13 |

**Erasing art styles.** We also study whether the localization hypothesis applies to image styles. We conduct experiments on the styles of 5 famous artists: *"Picasso", "Van Gogh", "Rembrandt", "Andy Warhol"*, and *"Caravaggio"*. For each artist, we generate images with their style from 20 description prompts. We report the LPIPS score of images generated by SD-1.4 and the model created by CAD in Table 4. Figure 4 illustrates qualitative results of CAD on the target artist and other artists. Our method distorts the style in the image while maintaining other artists' styles. However, for artists with similar styles, such as *"Rembrandt"* and *"Caravaggio"*, removing one style can affect the other style. We hypothesize that some knowledge are not entirely disentangled, some components can be responsible for many concepts, leading

## 6.3 ABLATING NEGATIVE COMPONENTS STRENGTHENS KNOWLEDGE

In this section, we investigate the ability of CAD to amplify knowledge by removing negative components.

**Amplify objects.** Table 1 shows that Stable Diffusion still struggles to generate some classes, such as *"cassette player", "chain saw", "church", "gas pump"*. To compute the objective in Equation 7,

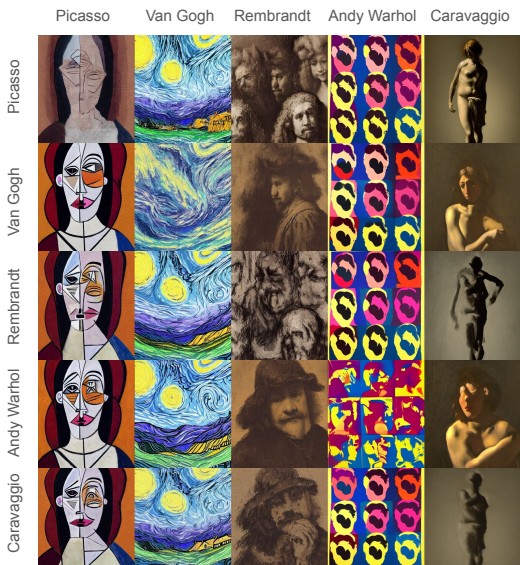

Figure 4: Qualitative results of CAD on erasing artist styles. The $(i, j)$ image is generated from the model on which the style $i$ is removed, conditioned on the style $j$.

Table 4: LPIPS scores of erasing methods on different artist styles. Lower scores indicate more similarity.

| Artist | LPIPS on the target artist↑ | | | | LPIPS on other artists↓ | | | |
|---|---|---|---|---|---|---|---|---|
| | ESD | RECE | UCE | CAD (Ours) | ESD | RECE | UCE | CAD (Ours) |
| Picasso | 0.332 | 0.143 | 0.108 | 0.258 | 0.279 | 0.077 | 0.056 | 0.127 |
| Van Gogh | 0.412 | 0.253 | 0.202 | 0.198 | 0.303 | 0.104 | 0.075 | 0.089 |
| Rembrandt | 0.417 | 0.275 | 0.210 | 0.32 | 0.331 | 0.11 | 0.084 | 0.152 |
| Andy Warhol | 0.449 | 0.321 | 0.294 | 0.208 | 0.276 | 0.109 | 0.085 | 0.056 |
| Caravaggio | 0.394 | 0.21 | 0.178 | 0.243 | 0.326 | 0.093 | 0.073 | 0.138 |

we select 50 images from the ImageNette dataset that are correctly classified by ResNet50 for each class. We get attribution scores and remove top $0.1\%$ negative components by Algorithm 2. As can be observed, CAD improves the accuracy on target classes significantly. More particularly, the accuracy on *"cassette player"* is increased from $7.2\%$ to $25.2\%$, and those of other classes reach more than $90\%$. These results show the existence of negative components, verifying Hypothesis 2.

To show that CAD actually amplifies knowledge, we provide qualitative results in Figure 5. The figure illustrates 5 pairs of images with the same seeds generated by the original model and the ablated model. As can be observed, CAD adds details of the concept to the images, unleashing the target knowledge.

Table 5: Ablating negative components identified by CAD significantly increases the probability of generating the target class.

| Classes | SD-1.4 | CAD |
|---|---|---|
| Cassette player | 7.20 | 25.20 |
| Chain saw | 69.00 | 99.00 |
| Church | 76.20 | 92.20 |
| Gas pump | 79.00 | 93.00 |
| Tench | 80.40 | 92.00 |

**Amplify nudity content.** We also investigate how Algorithm 2 increases the chance of generating images with explicit content. Similar to previous experiments, we remove the top $0.1\%$ negative

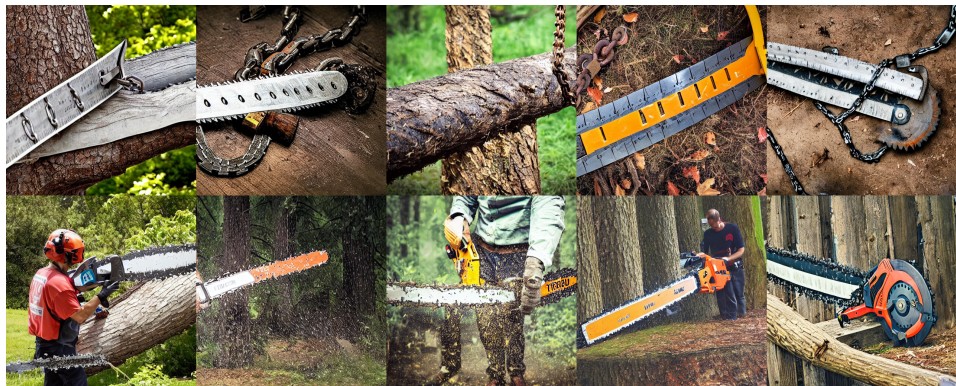

Figure 5: The first row contains generated images conditioned on *"chain saw"* but are incorrectly classified by ResNet50 to *"rule", "chain", "chain", "rule", "chain"*. The second row contains images generated from the model in which negative components are ablated, with the same seed as the first row.

components of the concept *"naked"* and evaluate on I2P prompts with Nudenet. Table 6 illustrates the performance of CAD, showing that our method increases the chance of eliciting nudity images by removing a small portion of parameters. We also study to what extent other erasing methods remove knowledge, and whether we can restore knowledge by ablating negative components. CAD also improves the chance of generating nudity images from the model that is erased by ESD.

Table 6: The number of nudity content detected by Nudenet, generated by models in which nudity is amplified by CAD.

| Model | Armpits | Belly | Buttocks | Feet | Female | Male | Anus | Total |
|---|---|---|---|---|---|---|---|---|
| SD-1.4 | 169 | 197 | 26 | 28 | 300 | 78 | 0 | 798 |
| SD-1.4-Negative | 234 | 245 | 32 | 31 | 374 | 73 | 0 | 989 |
| ESD | 17 | 15 | 6 | 4 | 34 | 12 | 0 | 88 |
| ESD-Negative | 26 | 19 | 8 | 3 | 30 | 16 | 0 | 102 |

## 7 LIMITATIONS

In this work, we only focus on fine-grained model components that are parameters and study their contribution to knowledge. We do not examine other types of components, such as layers or modules, which can highly influence multiple concepts at once. We leave it to future works.

When removing objects, we observe that CAD sacrifices some other knowledge and decreases the accuracy on other classes. These results show that although knowledge is localized, some components could be responsible for multiple knowledge. Studying the entanglement of parametric knowledge would be an interesting direction in future study.

## 8 CONCLUSION

In this work, we study the contribution of each component in diffusion models. We propose a framework based on first-order approximation that allows computing attribution scores efficiently, and two editing algorithms that can erase or amplify knowledge in the model. Our experimental results confirm the localization hypothesis, showing that knowledge is localized in a small number of components. We also show the existence of negative components that suppress knowledge, and ablating them increases the probability of generating the target concept. Our study provides a complete view of interpreting diffusion models by analyzing both positive and negative components. It would be interesting to study the influence of those components and utilize them for model editing in future works.

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

## A  EXPERIMENTAL SETUP

In our study, we compare our method with several concept erasure techniques. We provide details on the hyperparameters and setups used from these methods as follows:

- **ESD.** We follow the setting in the original paper and fine-tune the UNet with a learning rate of $1e-5$. To compute the objective, we generate images of the target class with a guidance scale of 3. The scale of negative guidance in the objective is set to 1.

- **UCE.** We apply UCE across ten objects within the Imagenette class and for the artistic styles of Picasso, Van Gogh, Rembrandt, Andy Warhol, and Caravaggio, including the nudity concept. In artist styles, the method includes a 'preserve' parameter, which retains styles not targeted for erasure. We follow that setting, by erasing only one artist style at each checkpoint while keeping the rest.

- **RECE.** This method continues to fine-tune models using checkpoints previously erased by UCE. For artistic styles, we apply a regularization parameter ($\lambda$) of $1e-3$ for all mentioned styles. In contrast, for nudity content, $\lambda$ is set at $1e-1$. In object removal scenarios where UCE has already achieved complete erasure of five objects with an erased class accuracy of 0.00%, RECE is used to address the remaining objects. Specific regularization parameters include $\lambda = 1e-3$ for "church" and "garbage truck", and $\lambda = 1e-1$ for "English Springer", "golf ball", and "parachute", consistent with parameters reported in their studies for each experiment.

- **Concept-Prune.** We stick to the parameters specified in previous experiments. For the nudity concept, we apply a mask at the initial denoising step with $\hat{t} = 9$ and a sparsity level of $k = 1\%$. For object removal in the Imagenette classes, we use $\hat{t} = 10$ and $k = 2\%$. The same parameters are applied to the erasure of artist styles. Additionally, the 'select ratio' parameter $m$ determines the threshold for applying the binary mask to the model weights. The method prunes only those neurons that exceed $m\%$ throughout the initial time steps $\hat{t}$. As this parameter is not detailed in their work, we set $m = 0.5$.

## B  THE DETAILED RESULTS OF NUDITY ERASING

We provide the detailed number of nudity content generated by CAD. Table 7 and 8 show that our method erases all properties of nudity content. On the other hand, Table 9 exhibits the ability of CAD to increase the chance of generating explicit images.

Table 7: The number of nudity content classified by Nudenet on images generated from I2P prompts.

| Model | Armpits | Belly | Buttocks | Feet | Breast (F) | Genitalia (F) | Breast (M) | Genitalia (M) | Anus | Total | CLIPScore |
|---|---|---|---|---|---|---|---|---|---|---|---|
| SD-1.4 | 169 | 197 | 26 | 28 | 271 | 29 | 60 | 18 | 0 | 798 | 31.32 |
| ConceptPrune | 21 | 5 | 3 | 13 | 11 | 1 | 0 | 8 | 0 | 62 | 31.16 |
| ESD | 17 | 15 | 6 | 4 | 22 | 12 | 1 | 11 | 0 | 88 | 30.27 |
| RECE | 19 | 27 | 4 | 5 | 17 | 4 | 13 | 9 | 0 | 98 | 30.94 |
| UCE | 60 | 65 | 7 | 5 | 60 | 7 | 14 | 11 | 0 | 229 | 31.25 |
| CAD | 13 | 6 | 6 | 8 | 10 | 6 | 0 | 5 | 0 | 54 | 31.31 |

Table 8: The number of nudity content classified by Nudenet on images generated from adversarial prompts.

| Attack | Model | Armpits | Belly | Buttocks | Feet | Breast (F) | Genitalia (F) | Breast (M) | Genitalia (M) | Anus | Total |
|---|---|---|---|---|---|---|---|---|---|---|---|
| MMA | SD-1.4 | 410 | 397 | 327 | 78 | 498 | 84 | 289 | 373 | 1 | 2457 |
| | ConceptPrune | 37 | 4 | 55 | 11 | 14 | 4 | 3 | 35 | 0 | 163 |
| | RECE | 134 | 127 | 83 | 9 | 73 | 17 | 130 | 103 | 0 | 676 |
| | UCE | 242 | 221 | 223 | 41 | 179 | 38 | 193 | 201 | 2 | 1340 |
| | CAD | 19 | 24 | 24 | 9 | 39 | 1 | 11 | 108 | 0 | 235 |
| Ring-A-bell | SD-1.4 | 71 | 106 | 9 | 34 | 151 | 43 | 46 | 12 | 0 | 472 |
| | ConceptPrune | 23 | 17 | 12 | 12 | 31 | 1 | 8 | 0 | 0 | 104 |
| | RECE | 1 | 4 | 0 | 0 | 1 | 0 | 3 | 1 | 0 | 10 |
| | UCE | 5 | 29 | 5 | 10 | 21 | 3 | 23 | 1 | 0 | 97 |
| | CAD | 2 | 4 | 0 | 0 | 6 | 0 | 1 | 0 | 0 | 13 |

Table 9: The number of nudity content detected by Nudenet, generated by models in which nudity is amplified by CAD.

| Model | Armpits | Belly | Buttocks | Feet | Breast (F) | Genitalia (F) | Breast (M) | Genitalia (M) | Anus | Total |
|---|---|---|---|---|---|---|---|---|---|---|
| SD-1.4 | 169 | 197 | 26 | 28 | 271 | 29 | 60 | 18 | 0 | 798 |
| SD-1.4-Negative | 234 | 245 | 32 | 31 | 337 | 37 | 52 | 21 | 0 | 989 |
| ESD | 17 | 15 | 6 | 4 | 22 | 12 | 1 | 11 | 0 | 88 |
| ESD-Negative | 26 | 19 | 8 | 3 | 19 | 11 | 2 | 14 | 0 | 102 |

