# OpenReview forum: "Unveiling Concept Attribution in Diffusion Models"
_ICLR.cc/2025/Conference — ICLR 2025 Conference Withdrawn Submission_

### Official Review · Reviewer_Dami · 2024-10-27

**Soundness:** 2
**Presentation:** 2
**Contribution:** 3
**Rating:** 3
**Confidence:** 4

**Summary:**

The authors propose an explainability method to interpret the weights of a diffusion model. In particular the method is meant to assess which weights contribute to a certain concept. They demonstrate applications of concept erasing and localizations. They provide experiments to demonstrate the quality of the interpretability method and visual results for editing.

**Strengths:**

1. The proposed method was able to remove weights while keeping the model performance for allowed outputs.
2. The proposed method demonstrate superior qualitative results for removing nudity.
3. The authors chose a challenging task, as the autoregressive nature of diffusion models prevents comparing final generation results to some ground truths or input them to classifiers and train. The general approach they take is elegant: Comparing one step results to one step results with a different condition.

**Weaknesses:**

## 1) Method:
As far as I understand, the method boils down to grant importance to each weight by looking at the derivative of the objective w.r.t. that weight (scaled by the weight itself, which is equivalent to normalizing i.e. instead of quantifying the change in w, quantifying the change in percents of w). This exposes several weaknesses:

1. __Limited novelty__: quantifying importance by gradients has been used in many cases. Sometimes w.r.t. inputs, sometimes with weights. applying it in the particular case of diffusion models weights is not novel enough for this venue, to my taste.

2. __Another novelty point__: This is almost the first part of simple fine-tuning. Fine-tuning would be to take this score (but unnormalized, this is why I wrote "almost"), multiply by some learning-rate and subtract from the weight. This almost feels like gradient descent without the descent.

3. __Quality of the method__: I think this measure is a too loose approximation. While expressing the objective as a linear function of the weights is reasonable, it is important to make the distinction: determining the importance of each weight in this calculation according to the derivative w.r.t. it is different, an I believe- too local. It is equivalent to assuming that the model itself is linear. It is far less accurate than the mentioned Shah et al.- By randomly zeroing weights and learning the effect you get much better assessment of the importance as it is not local. To make this claim clearer, here is an example. say the objective as a function of one of the weights saturates, e.g., the function is Sigmoid. Around high values and values close to zero for this weight this measure will be almost zero. However for mid-values it can be high. An extreme case is a shifted heavy-side (step function), so that Alpha=1 for x>3 and Alpha = 0 for x<=3. If the relation is close to it, it means that around 3 the derivative is extremely high, but zero in all places. It is not uncommon for weights to act somewhat binary in some range, especially with ReLUs around. In such a case, Shah et al. would perform reasonably, as there will be reasonable difference between turning this weight "on" and "off". Shah et al. also makes more sense as it corresponds with the applications. The applications zeroize weights, just like in Shah's measure. They don't change them by epsilon. It could be that the authors could have gotten better results if they took several gradient steps instead of zeroizing these weights.

## 2) Experiments:
1. __Figure 1__: I wouldn't say Pearson correlation of 3.5 counts as high, but actually weak to moderate. Also examining the figure visually, I think it looks only slightly correlated. I think this highlights the weakness of the the proposed approach rather than its success.

2. __Qualitative results__:  In general, I think the results are not in par with recent editing methods. The results by the ablated model don't have any content. I would have been impressed, if the entire image mostly would remain the same except for the concept. At the very least, I would expect the final image would look like something realistic. The visual results for removing concepts generate images with no content. Actually in some cases, there is even still a trace of an inappropriate result (e.g., fig3 rightmost)- This implies that it is not the concept that was deleted. One could claim that it is a reasonable goal to detect inappropriate prompts and just not output anything reasonable. However, in such a case all you have to do is detect inappropriate prompts and block them. It would have made sense if the authors would have demonstrated cases where a regular prompt yields inappropriate image that can be detected and blocked efficiently.

3. __No baselines__: For assessing the quality of the interpretability mechanism, I would expect some reasonable baselines that show that indeed this measure has a distinctive meaning: Most basic is removing random weights. But also doing some gradient steps instead of zeroizing weights (as mentioned in 1.3). Also in Figs 2 and 3, a comparison of the allowed prompt of the ablated model with the non ablated model is missing. If the image changes drastically between them it shows that the process was far from cleanly removing the concept.


## 3) Readability and strength of claims:
1. __"Editing" is over-claiming__: I would not describe the experiments as applications. I would describe them as ways to showcase the quality of the interpretability method. There are very strong editing methods and I don't think this should compete with them.

2. __Not defining the objective__: J is only defined for the case of concept removing. There is no definition of it I could find before that. If it's always that J then have it defined early on- it is the most important value dealt with. If there are many possible Js, then have an explanation early on that J could be many things and provide some examples.

**Questions:**

I appreciate the thought and direction taken by the authors, but I think this paper is not ready and my main suggestion is to improve the method.

---

### Official Review · Reviewer_FhKK · 2024-10-27

**Soundness:** 3
**Presentation:** 2
**Contribution:** 2
**Rating:** 5
**Confidence:** 4

**Summary:**

Given a concept, the work aims to find neuron/parameter attributions in diffusion models. Their proposed method, CAD, approximates the attribution scores for a parameter by using a linear model. They also utilize Taylor-approximations for computational efficiency in calculating the scores (in comparison to brute-force multiple score calculations for each parameter). They find that CAD can estimate both positive and negative attributions for a concept inside diffusion models. Ablating the positive attributed neurons leads to competitive erasure while ablating negative attributed neurons leads to enhancement of the concept.

**Strengths:**

The application of attribution neurons in concept erasure is very interesting! Especially with the strong quantitative results competing with the state-of-the-art methods is very convincing.

The introduction of positive and negative attributions is quite intriguing!

**Weaknesses:**

1. The technical novelty seems scarce, in the current context. The authors may add more context in the manuscript to make their technical contribution more convincing.

At first glance, the method looks very similar to
a. "Attribution patching" and "Edge attribution patching" (https://openreview.net/pdf?id=tiLbFR4bJW) from Language models. Where they propose similar attribution methods using Taylor series expansion and arrive at very similar attribution score function.
b. The authors may also refer to "Cones: Concept Neurons in Diffusion Models for Customized Generation" (https://proceedings.mlr.press/v202/liu23j/liu23j.pdf), where they found similar attribution neurons in the context of customization. In fact they do not make linear assumptions and still arrive at the w.(dL/dw) corresponds to attribution.


2. The methodology needs more explanation: What parameters did the authors calculate the scores on (is it just the cross attentions? Or all the layers?).

3. The qualitative example for art-erasure seems to indicate that the model doesn't completely erase the knowledge from the models.

**Questions:**

1. What neurons/parameters does CAD calculate the attribution scores for? And what does the distribution of these neurons look like for a few concepts?
2. When CAD calculates the attribution for erasure using the objective from "Ablating Concepts" Kumari et al. - what is the attribution being calculated on? Is it the concept attribution or attribution that makes the diffusion model output some other concept (" ") when prompted for the concept. It almost seems like the second case - I am curious as to what the authors think
3. Instead of ablating the neurons during erasure - what happens when you run the prior erasure technique (ESD or CA) to only fine-tune the attributed neurons? does it perform well in general compared to fine-tuning all the layers? This may show how important concept neurons really are in the context of using it for any up-stream tasks
4. I would appreciate authors views on how CAD complements attribution methods from language models like Syed et al. https://openreview.net/pdf?id=tiLbFR4bJW  and how CAD's attribution connects to the observations made in Cones work https://proceedings.mlr.press/v202/liu23j/liu23j.pdf

---

### Official Review · Reviewer_M7qv · 2024-10-28

**Soundness:** 2
**Presentation:** 2
**Contribution:** 2
**Rating:** 3
**Confidence:** 3

**Summary:**

This paper analyzes how individual components in diffusion models contribute to representing specific concepts, such as objects or styles. It further extends this finding to model editing by removing components that contribute to a target concept, effectively erasing specific objects or styles.

**Strengths:**

Overall, the author explains the methodology and motivation clearly.

**Weaknesses:**

****Concerns on Methodology and Hypotheses****

Overall, the methodology is based on two very strong hypotheses, but the experiments do not fully persuade me that these hypotheses are valid.

For example,  _“Knowledge is localized in a small number of components. If we remove those components associated with a concept c, the model will not generate c, but other concepts will not be affected.”_ However, styles and objects represent fundamentally different types of knowledge; is it correct to treat them in the same way? In Figure 4, I am not convinced that the style has been successfully erased, as some styles (e.g., Van Gogh) remain quite obvious, and the image is not “attacked.”

Also, how can you be sure that when you remove a concept, you will not also ablate other unwanted knowledge if you arbitrarily set the corresponding components to zero? According to the pigeonhole principle, there are countless concepts in the world, yet you only have 860 million parameters in SD-1.4. As you mentioned in the limitations section, a single parameter might contribute to several concepts, and different concept and parameters may be intertwined. I am not convinced that setting a parameter to zero is an effective method for removing a concept. Or at least not from the way you find the parameters that contributes to a concept

**General Image Quality of ablated model after concept removal**

Maintaining general Image quality after your modification is also crucial. How can you ensure that the quality of generated images remains at a certain level after your modifications? For example, is the FID score of your model maintained (or not significantly decreased) compared to SD-1.4? From the example images, it appears that the quality of generation has dropped considerably; for instance, a simple prompt like "a few boats out on a river" still does not generate accurately. Additionally, a prompt such as "a photo of a church" yielded many grayscale images, which typically suggests a decline in image generation quality in diffusion models

**Writing mistakes and suggestions**
- Line 425 contains an incomplete sentence.
- Figures: Some figures use boldface to highlight the best scores, others use boldface plus an underscore, and some do not use any highlighting. This inconsistency could be misleading or confusing for readers, so ensuring a consistent style across all figures would improve clarity.

**Questions:**

- Are there any experiments to validate that the image generation quality of your model doesn't decrease significantly with your modification?
- questions in the weakness section - Concerns on Methodology and Hypotheses.

---

### Official Review · Reviewer_Wyfj · 2024-11-04

**Soundness:** 3
**Presentation:** 2
**Contribution:** 1
**Rating:** 5
**Confidence:** 5

**Summary:**

This paper explores how different components in a diffusion model contribute to the generation of a particular concept. The key idea is to use gradient-based attribution techniques to identify the components that contribute most (either positive or negative) to a concept. After the concept-related components are identified, they can be used to remove or amplify the generation of concepts.

Two sets of experiments are conducted. For concept removal, the paper studies the removal of objects, nudity, adversarial prompts, and styles. For concept amplification, the paper focuses on the amplification of objects and nudity. In both sets of experiments, the proposed method is comparable or outperforms existing baselines.

**Strengths:**

- Concept attribution is an important problem, which can be useful for mitigating safety/copyright/privacy challenges in image generation.
- The idea of finding negative contribution is interesting, allowing for concept amplification, which seems a novel application to me.
- Experiments show the effectiveness of the proposed method on many concepts.

**Weaknesses:**

My main concern of this paper is its similarity to existing works on gradient-based attribution / attribution patching [1] [2] [3] [4]. The "key idea" of the proposed method is to use first-order Taylor expansion for the loss function. However, this idea is also the core of gradient-based attribution / attribution patching. It seems to me the novelty lies in the application of gradient-based attribution to diffusion models, but not the method itself.

[1] Self-Attention Attribution: Interpreting Information Interactions Inside Transformer https://arxiv.org/abs/2004.11207

[2] Attribution Patching Outperforms Automated Circuit Discovery https://arxiv.org/abs/2310.10348

[3] Attribution Patching: Activation Patching At Industrial Scale https://www.neelnanda.io/mechanistic-interpretability/attribution-patching

[4] Axiomatic Attribution for Deep Networks https://arxiv.org/abs/1703.01365

**Questions:**

How is the proposed method different from gradient-based attribution / attribution patching?

---

### Note · Authors · 2024-11-15

I have read and agree with the venue's withdrawal policy on behalf of myself and my co-authors.